# A comparative test of inequity aversion in domestic dogs (*Canis familiaris*) and dingoes (*Canis dingo*)

Katherine McAuliffe[1,2]*

**1** Department of Psychology and Neuroscience, Boston College, Chestnut Hill, Massachusetts, United States of America, **2** Department of Human Evolutionary Biology, Harvard University, Cambridge, Massachusetts, United States of America

* katherine.mcauliffe.2@bc.edu

## Abstract

Despite much recent empirical work on inequity aversion in nonhuman species, many questions remain about its distribution across taxa and the factors that shape its evolution and expression. Past work suggests that domestic dogs (*Canis familiaris*) and wolves (*Canis lupus*) are averse to inequitable resource distributions in contexts that call upon some degree of training such as 'give paw' and 'buzzer press' tasks. However, it is unclear whether inequity aversion appears in other canid species and in other experimental contexts. Using a novel inequity aversion task that does not require specific training, this study helps address these gaps by investigating inequity aversion in domestic dogs and a closely related but non-domesticated canid, the dingo (*Canis dingo*). Subjects were presented with equal and unequal reward distributions and given the opportunity to approach or refuse to approach allocations. Measures of interest were (1) subjects' refusal to approach when getting no food; (2) approach latency; and (3) social referencing. None of these measures differed systematically across the inequity condition and control conditions in either dogs or dingoes. These findings add to the growing literature on inequity aversion in canids, providing data from a new species and a new experimental context. Additionally, they raise questions about the experimental features that must be in place for inequity aversion to appear in canids.

## Introduction

Cooperation, where one individual provides a benefit to another individual [1], is vulnerable to exploitation by agents who benefit from collective action without investing in it. One way in which humans may avoid such exploitation is by paying close attention to how the spoils of collective action are distributed and punishing individuals who take more than their fair share [2–5]. Human adults exhibit a strong aversion to unfair outcomes and are willing to sacrifice personal resources to avoid receiving relatively less than a peer (disadvantageous inequity; [6–9]) and, in some cases, more than a peer (advantageous inequity; [7, 9]). This aversion to

**Data Availability Statement:** Data and R code are in the Supporting information files.

**Funding:** The author received a graduate research grant from the department of Human Evolutionary Biology at Harvard University to support this work.

**Competing interests:** The author has declared that no competing interests exist.

unequal outcomes—so called *inequity aversion*—is considered to be an important psychological mechanism that contributes to the maintenance of cooperation among unrelated individuals [4, 10, 11].

While inequity aversion is thought to help regulate human cooperation, the extent to which it is involved in nonhuman animal (hereafter 'animal') cooperation remains unclear. Over recent years, a number of studies have started to address this question by testing for inequity aversion in animals, with a particular focus on primate species. Brosnan and de Waal [12] conducted the first experimental test of inequity aversion in a nonhuman primate by testing whether brown capuchin monkeys (*Sapajus* [*Cebus*] *apella*) would refuse to participate in a token trading task in which they received a lower payoff than a partner for trading the same token. Results showed that subjects were less likely to trade tokens in the *inequity* condition, wherein the subject received a less desirable reward than her partner, than in a condition in which both monkeys could trade for equal rewards. These findings were interpreted as the first evidence for inequity aversion in a nonhuman species and suggested that inequity aversion has deep evolutionary roots.

A number of studies with nonhuman primates have built on Brosnan and de Waal's original experiment. Some have provided evidence for inequity aversion (e.g., capuchins: [13, 14] and chimpanzees (*Pan troglodytes*) [15]) whereas others have provided no evidence for inequity aversion (e.g., capuchins [16–20]; chimpanzees [21, 22]). Given the conflicting results, it remains difficult to assess the extent to which inequity aversion drives decision-making in the domain of cooperation in primates. Consequently, evidence from non-primate species would be helpful in probing the purported relationship between cooperation and inequity aversion in animals.

To this end, several recent studies have tested inequity aversion in the domestic dog (hereafter 'dogs', *Canis familiaris*; reviewed in [23]), a non-primate species that shows extensive intra- and inter-species cooperation (reviewed in [24]). In the first test of inequity aversion in dogs, Range and colleagues [25] tested pairs of companion dogs on a version of Brosnan and de Waal's [12] inequity aversion experiment, except that instead of trading tokens, subjects were asked to 'give paw' to an experimenter in order to receive a reward. Participants in this study were trained to 'give paw' prior to their participation. Further, a pre-test assessment session was conducted to ensure that dogs would reliably respond to this command before their inclusion in the study. Range and colleagues found that dogs refused to give paw more in a *reward inequity* condition, in which the subject received no reward while their partner was rewarded, compared to an *equality* condition and controls [25].

This initial study using the 'give paw' task catalyzed a series of follow-up studies on inequity aversion in dogs. For instance, Range et al. [26] explored whether the individual-level factors such as motivation as well as the quality of the relationship between the two subject dogs affected their responses to inequality. Brucks and colleagues [27] replicated the original finding that dogs were sensitive to reward inequality and additionally showed that exposure to inequity affects tolerance in a subsequent food-sharing task. In a separate study, Brucks and colleagues [28] demonstrated a link between inhibitory control and inequity aversion: dogs with slower decision speeds—i.e., less reactive dogs—showed a stronger response to inequity as measured by their refusal to continue participating in the 'give paw' task. Finally, McGetrick et al. [29] again replicated the inequity aversion response in the 'give paw' context yet found no difference between cooperative and non-cooperative breeds, the former of which were hypothesized to show a stronger response to inequity. Taken together, studies using the 'give paw' task suggest that dogs reliably show sensitivity to receiving no reward when a familiar partner is receiving a reward for performing the same action.

Outside of the 'give paw' context, inequity aversion in dogs has been studied in a choice paradigm. Horowitz [30] tested dogs' preferences for fair versus unfair trainers. In this task, a participant dog was paired with a confederate and habituated to (1) a fair trainer (who rewarded dogs equally) and a disadvantageously unfair trainer (over-rewarding trainer: more for confederate, less for subject) or (2) a fair trainer and an advantageously unfair trainer (under-rewarding trainer: more for subject, less for confederate). Dogs showed a preference for the over-rewarding trainer compared to the fair trainer but did not prefer the fair trainer to the under-rewarding trainer. These results suggest that dogs can maximize their own payoff relative to available payoffs, but they do not provide clear evidence that dogs avoid unfair experimenters. Moreover, it is unclear whether the confederate partner played an important role in guiding the subjects' decisions because non-social controls (i.e., where the payoff distribution was unequal but no partner was present) were not conducted. However, because dogs were not required to perform a task in exchange for a reward—an experimental feature thought to be important in inequity aversion tasks—results from Horowitz may not present a major challenge to the claim that dogs are averse to inequity (see [23] for further arguments).

The finding that dogs are averse to situations in which they exert the same effort as a partner yet receive a relatively lower payoff is broadly consistent with the idea that dogs exhibit mechanisms for avoiding an unfair share of the spoils of collective action. However, inequity aversion in dogs is also consistent with an important alternative account. Namely, domestic dogs may be averse to inequity because they have evolved sensitivity to how food is distributed from humans. Such sensitivity may have arisen since domestic dogs receive virtually all their food from humans and are frequently provided with food rewards as payment for work. This account suggests that inequity aversion in domestic dogs is a product of selective pressures during the species' history of domestication (artificial selection or by-product of selection on other traits). Specifically, throughout domestication, it may have been advantageous for dogs to (1) pay attention to how food rewards are distributed by humans and (2) ensure that they were rewarded at least as well as their counterparts. Critically, this hypothesis predicts that inequity aversion will be observed in domestic dogs but will be absent in other canid species.

Recent work provides evidence against this account. Essler and colleagues [31] tested pack-living dogs and pack-living wolves on a 'buzzer press' task. Like, the 'give paw' tasks, participants in this study were trained to perform an action—in this case, to press a button with their paw—in exchange for a reward. Results from this study showed *both* dogs and wolves were averse to situations in which they received no reward for pressing the buzzer while their partner received a reward. The result that wolves showed inequity aversion in this paradigm lends support to the claim that inequity aversion is not a product of domestication and, instead, may be deeply rooted in the social canid lineage. Interestingly, this same task has not yielded inequity aversion in non-pack-living dogs, a finding which is, at present, poorly understood but is under active investigation [32].

Although past work on dogs and wolves suggests that at least some social canids are averse to inequity—a response that may be tied to cooperation and the challenges that arise from it (e.g., food sharing, cooperative hunting, cooperative defense)—it also raises important questions. First, do we see inequity aversion in other non-domesticated social canid species? Second, do we see inequity aversion in other contexts, including those that do not require training prior to the inequity task? This study begins to address these outstanding questions by (1) testing inequity aversion in a non-domesticated social canid in addition to domestic dogs and (2) employing a novel task that requires no specific training and, as in [30], is based simply on approach behavior.

First, to test whether inequity aversion is observed in other canids, this study examined inequity aversion in Australian dingoes (*Canis dingo*). Dingoes are an ideal comparative

species because they are closely related to dogs [33–36] and show intraspecific cooperation in the wild. Dingoes live in family groups [37, 38], cooperatively rear their young [39] and cooperatively hunt prey [40]. Extant dingoes are not considered to be a domesticated species [24, 37, 41]. Although the extent to which dingoes ever were domesticated is a controversial subject (see [42] for a review), it is generally accepted that dingoes differ markedly from dogs in their behavior, physiology and cognition [37, 42]. Thus, dingoes offer a unique opportunity to study inequity aversion in a closely related but non-domesticated cooperative species. Moreover, the dingoes tested in this study have extensive experience with humans and are fed daily by humans, attenuating the possibility that any observed differences between dogs and dingoes are due to differential exposure to humans and human feeding.

Second, the current study employed a new task that required no specific training and was instead based on subjects' approach behavior. An experimenter distributed food between two plates and subjects then had the opportunity to approach the plates. Subjects thus had visual access to both their plate and their partners' plate prior to making a decision about whether to approach. In this respect, the current method was similar to methods of resource distribution used in studies of human adults [8] and children [43–45] as well as in recent work on inequity aversion in other species [17, 46, 47]. The main prediction in this study is that if subjects are sensitive to inequity then they should be less motivated to approach under conditions of reward inequity (when they receive nothing while their partner receives a reward) than under control conditions. Subjects are expected to stop approaching eventually across conditions because they were not reinforced. However, they were expected to stop *earlier* in the context of reward inequity than in the various control conditions.

The present study builds on the methodology and findings of past work [25, 26, 30] but focuses exclusively on *reward inequity*: cases in which a subject receives no reward while their partner receives a reward. In an effort to control for potential relationship effects existing between familiar dogs, subjects were paired with a confederate partner as opposed to a familiar partner. In this way, the present study again aligns with how inequity aversion is studied in human adults and children (e.g., [43, 48]).

This study examined three measures of interest. The first measure was subjects' motivation to approach an unequal reward allocation under different experimental conditions or, more specifically, their *lack* of motivation to approach in the absence of rewards. The second measure was subjects' *latency* to approach (i.e. reaction time) across conditions. Reaction time is a valuable, complementary measure to subjects' approach behavior because it can provide additional insight into subjects' motivation. If dogs or dingoes show inequity aversion, they should be less motivated—and thus slower—to approach when they are receiving a disadvantageous payoff (see [46], for a latency measure in an inequity aversion task with fish). The third measure was subjects' tendency to look at (i.e. "reference") humans in different experimental conditions (e.g., [42, 49]). The prediction was that dogs would be more likely than dingoes to reference humans and that, across species, social referencing would be highest when subjects were presented with a disadvantageously unequal payoff. Social referencing was included as a measure because, in line with work using unsolvable tasks with canids (e.g., [42, 49]), subjects may treat unequal payoffs as a 'problem' that needs to be solved, in which case one would expect to see more social referencing in the experimental than control conditions.

## Method

### Subjects

Subjects were 72 dogs (47 females, 25 males) and 11 dingoes (8 females, 3 males). This difference in sample size was due to differences in availability of subjects of each species. Dogs were

tested in a between-subject design while dingoes were tested in a within-subject design (see below for details).

Dogs were recruited and tested at a Canine Lab in the USA during October 2010 –May 2011 (see S1 Table in S1 File for subjects by sex and breed). Dogs were brought to the lab by a human companion who remained with them for the duration of testing and handled them at all times. A requirement for inclusion in the task was that dogs were brought to the lab on two separate days (see below for details). Data from dogs brought to the lab on only one day were not included. Dingoes were tested at the Dingo Discovery and Research Centre in Toolern Vale, Victoria, Australia during July and August 2010 (see S2 Table in S1 File for subjects by sex). Dingoes were selected as subjects if they did not exhibit anxious behaviors when approached by the experimenter (e.g. alarm barking, hiding). Dingoes were housed in pairs in pens of 30 square meters, with indoor/outdoor access. All dingoes were given regular access to exercise enclosures ranging from 400 square meters to 1500 square meters. Water was piped to automatic dispensers in each night pen, and to each exercise enclosure. Dingoes were fed a combination of high-quality commercial dog food and meat once daily. Staff members at the Dingo Discovery and Research Centre and a research assistant handled the dingoes in all experiments.

Both dogs and dingoes were tested with an unfamiliar confederate partner (hereafter 'partner'). Partners for dog subjects were two dogs from the local community. During testing, partners were handled by research assistants. Before testing, the subject dog and the partner were allowed to greet one another while on leashes to minimize the risk of aggression during the experiment. The dingo partner was a female dingo living at the Sanctuary who was chosen to be the partner a priori because she had not socialized with the other subjects. Immediately prior to dingo testing, the subject was walked past the partner either in her home enclosure or in the testing arena, depending on which individual was brought into the testing area first. For dingo subjects, a single experimenter conducted all sessions. For dog subjects, six different experimenters were involved in the study. Experimenter remained consistent across both sessions for dogs.

Food rewards for both dogs and dingoes consisted of small meat-flavored treats. The majority of dogs were given small pieces of sausage (Natural Balance™ Dog Food Roll, beef formula). In cases where a subject had a food allergy or refused to eat sausage, they were tested using Zuke's™ Mini Naturals Dog Treats or the owner's treats. Dingoes were tested using small pieces of raw meat (either pieces of beef tip or a small (quarter sized) dollop of minced meat). All food pieces were approximately the same size. Food rewards were present during all sessions, even during conditions where no food was distributed, to control for the possibility that the presence of food rewards could affect subjects' behavior.

## Experimental set-up

Dogs were tested in a laboratory room (Fig 1). Prior to the experiment, the subject was brought into the testing area and given a few minutes to explore and habituate to the testing room. Once exploratory behavior ceased, the partner was brought into the testing room and both dogs were attached to leashes that were connected to two chairs in opposite corners of the room. The subject and partner handlers were instructed to sit in the chairs and hold the dogs' leashes until the experimenter indicated that the dogs could be released (by signaling with a clear drop of the head). A large Plexiglas barrier separated the dogs. The experimenter stood at the barrier and placed food on two plates that were positioned on either side of the barrier. On the subject's side of the barrier, a large rectangular "approach box" was marked on the floor by black tape (roughly 4.5 x 3.5 feet).

(A)

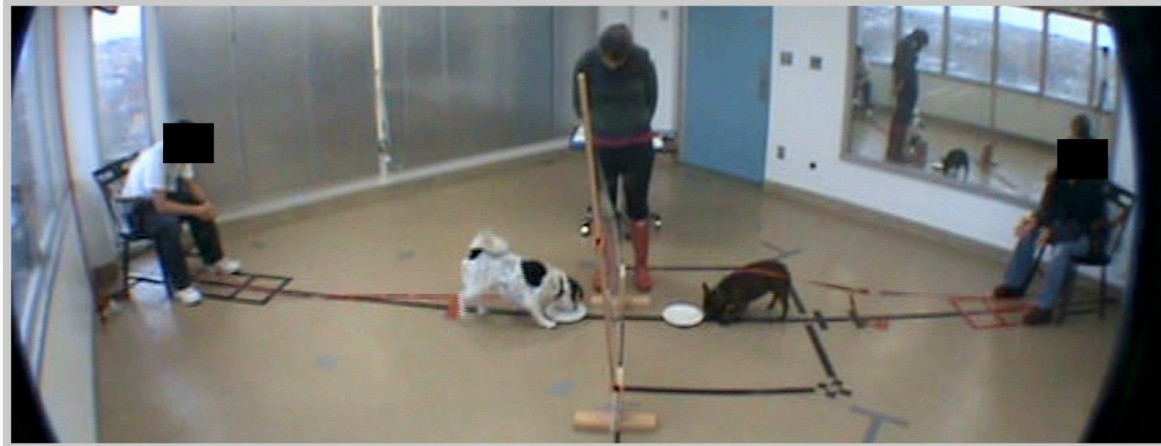

(B)

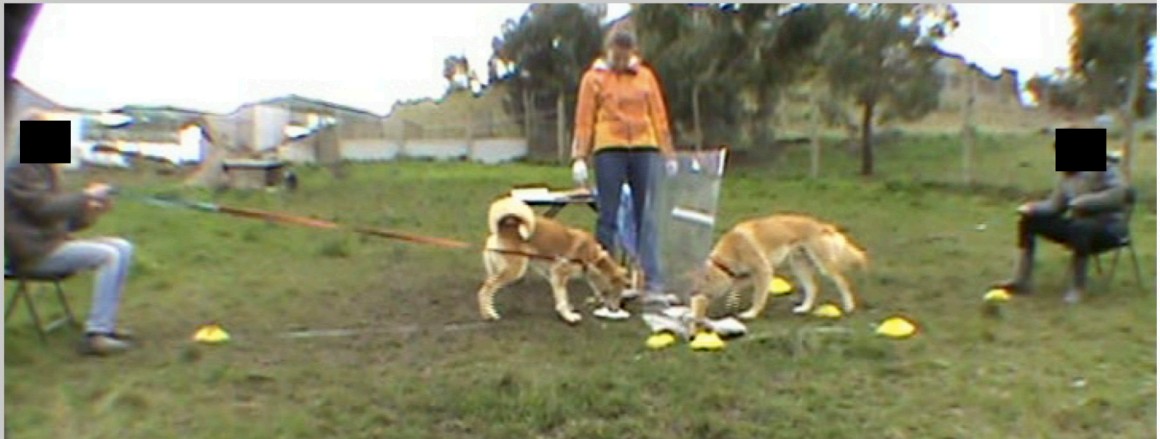

**Fig 1. Testing set-up for dogs (A) and dingoes (B).** The subject's handler and subject sat on the right of a transparent barrier and the partner and the partner's handler sat on the left of the barrier. The experimenter stood in the middle of the testing space, called the subjects, and distributed rewards on plates. A subject was considered to have approached when he/she had two front paws in the box surrounding the plate (box marked by black tape for dogs and yellow cones and blue ribbon for dingoes).

The experimental set-up for the dingoes was equivalent to the dog set-up with the major difference being that dingoes were tested in a fenced outdoor enclosure rather than in a laboratory (Fig 1). Dingoes were tested outside because the majority of dingo subjects were not accustomed to being indoors. All dimensions of the testing area were identical for dog and dingo subjects but the approach box for the dingoes was marked on the ground by blue ribbon and yellow cones instead of black tape. This difference in marking was necessary because tape would not stick to the grass.

## Research design

Before testing, subjects were given warm-up trials to ensure that they were comfortable approaching and eating from the plates. During warm-ups, both subject and partner were called to the plates by the experimenter at the same time and given a reward (see procedure below). Subjects had to approach on two consecutive warm-up trials before they proceeded to the experiment.

**Table 1. Description of conditions administered in the test half of sessions.** Prior to the test half, subjects were tested in a baseline condition where, with one exception, both subject and partner were rewarded for approaching the plate. In the Non-Social Control 2 (NSC2) condition, the baseline was run in the absence of a partner dog and thus food was simply placed on the partner's plate. Food was always absent from subjects' plates during tests. A single unit of reward (note that reward type varied between dogs and dingoes) was used in all cases.

| | Condition | Description | Partner present during test? | Food on partner's plate? |
|---|---|---|---|---|
| Baseline | Social | Subject and partner are called to plates and both individuals are rewarded for approaching. This baseline was run before all conditions except NSC2. | N/A | N/A |
| | Non-social | Partner is absent. Subject is called to plate and is rewarded for approaching. Food is placed on both plates. This baseline was run before NSC2. | N/A | N/A |
| Social | Inequity | Subject and partner are called to plates and only the partner is rewarded for approaching. | Yes | Yes |
| | Social No Reward (SNR) | Subject and partner are called to plates and neither is rewarded for approaching | Yes | No |
| Non-Social | Non-Social Control 1 (NSC1) | Subject is tested without partner and is called to plate but is not rewarded for approaching. Food is, however, placed on the partner's plate. | No | Yes |
| | Non-Social Control 2 (NSC2) | Same as NSC1 except that subject is tested in the baseline without a partner. | No | Yes |
| | Non-Social No Reward (NSNR) | Subject is tested without partner and is called to plate but is not rewarded for approaching. No food is placed on the partner's plate. | No | No |

After warm-up trials the experimental session began. Only one session was run per day. Each experimental session consisted of two parts. The first part of each session consisted of 15 "baseline" trials in which both subject and partner were given a food reward for approaching the plates. Following baseline trials, subjects were given a short break and were offered water. In one case during dingo testing the water break was skipped due to inclement weather. Following the short break, 15 test trials were conducted. The distribution of rewards in test trials varied depending on condition as did whether or not the partner was present (see Table 1 for a description of conditions). Regardless of condition, subjects never received a treat for approaching the plate during test trials. Test trials were always preceded by baseline trials in order to control for the possibility that subjects would lose motivation over the course of a session: because the non-rewarded part of the session always occurred after the rewarded part of the session, the potential effects of loss of motivation were consistent across all conditions.

The main condition of interest was the *Inequity condition*. In this condition, both subject and partner were present and the partner received a reward for approaching but the subject did not receive a reward for approaching. In addition to the Inequity condition, four different control conditions were administered.

As in Range et al. (2009), a *Social No Reward condition* (SNR) was conducted. In this condition, both the subject and partner were present but neither was rewarded for approaching. In order to ensure that the partner approached during every trial, even on trials where no food had been placed on his/her plate, the experimenter secretly (out of sight of the subject) gave the partner a treat at the end of each SNR trial, during the return to his handler. This condition was conducted in order to control for the possibility that subjects may stop approaching in the Inequity condition because of frustration due to lack of food, which may be compounded by distraction by their partner (thus partner presence was held constant between this condition and the Inequity condition). In addition to this social control, two non-social controls were conducted.

During both the *First* and *Second Non-social Control condition* (NSC1; NSC2), the partner was absent but food was placed on the partner's plate. The only difference between these conditions was that during NSC2, the partner was absent during baseline trials as well as during test trials and thus it was a completely non-social session. The logic of NSC2 was that it allowed

for the assessment of how subjects would behave in a completely non-social version of the task.

Finally, to understand whether subjects would be motivated to participate in the task in the absence of a partner and in the absence of a reward, a *Non-social No Reward condition* (NSNR) was run. In this condition, the partner was absent and no food was placed on either plate.

Subjects were predicted to show lower levels of approach behavior in all five experimental conditions compared to the baseline, when they were reinforced for approaching. However, if —as in [25]—subjects are particularly averse to situations in which their partner is being rewarded for performing an action while they are not being rewarded, they should cease participation earlier in the Inequity condition than in the controls.

All five experimental conditions were run with both dog and dingo subjects. However, dogs were tested in a between-subject design while dingoes were tested in a within-subject design. This difference was due to the fact that it would have been burdensome for dog owners to bring their dogs to the lab on five different days. Consequently, each dog was tested on two different days and was tested in the Inequity condition and one of the four control conditions. Thus, dog subjects could be grouped into four different *control groups* (SNR = 19 dogs; NSC1 = 18 dogs; NSC2 = 16 dogs; NSNR = 19 dogs). In each control group subjects were tested in the Inequity session and the associated control session. The order in which the inequity versus control sessions was administered was counterbalanced (S1 Table in S1 File). Dingo subjects, on the other hand, were each tested in all five conditions on five separate days. The order in which conditions were administered was counterbalanced between dingoes using a Latin square (S2 Table in S1 File).

## Procedure

The experimental procedure was identical for dog and dingo subjects. In all trials, the experimenter held up two pieces of food, called the subject and partner by using the command "Come [subject/partner name]!" The order in which names were called was counterbalanced within session. The experimenter then bent down and simultaneously placed a piece of food on each of two plates (or simultaneously touched the food to the plates in conditions where food was not being distributed), stood up, and signaled (by dropping his/her head) to the handlers that they could release the subject and partner. Subjects were then given approximately five seconds to approach (mean approach interval time for dogs: 4.37 ± 1.33 seconds (s), mean approach interval time for dingoes: 5.03 ± 1.81s). The approach distance from the release area to the plate was approximately six feet. After the subject and partner approached, the experimenter returned the subject to his/her handler and then returned the partner to his/her handler. Subjects had an opportunity to eat the food item from the time they approached to the plate until their next approach. However, since food items were small, they were typically eaten immediately. Between trials, experimenters made live coding notes and gathered food items for the next trial, activities which typically took a total of around 10 to 30 seconds. If a subject did not approach on a given trial, the experimenter asked the handler to hold the leash and made a motion to return the dog in order to keep all movements during the experiment consistent regardless of subject behavior. Similarly, in conditions where the partner was absent, the experimenter continued to pretend to return the absent partner to the partner handler (who was always present). Food that was not placed on a plate was held in the experimenters' hands and returned to the table located behind them which held the bowl of food along with the live coding sheets.

If a subject did not approach for five consecutive trials, the session was stopped to avoid causing unnecessary stress to the subjects. This never happened during the baseline half of

sessions but happened regularly during the test half of sessions (28% of dog test sessions; 51% of dingo test sessions). In cases where a trial was deemed invalid during testing (e.g., due to a mistimed dog release by the handler) the trial was redone (Dogs: 24 trials were deleted and replaced with extra trials for issues related to attention (4); experimenter error (1) and handler error (19); Dingoes: 14 trials were deleted and replaced with extra trials for issues related to attention (11); experimenter error (1) and handler error (2)). Extra trials that were erroneously run were not included in analyses. Two dog subjects were excluded due to issues with at least one of their visits (reluctance to approach when getting food (1); experimenter error (1)).

## Video coding and dependent variables

All sessions were recorded and research assistants coded data from video recordings. Videos were coded for several different measures. First, trials were inspected for experimental error. To this end, coders ensured that on all trials the food was presented in the dog's field of view ("attention"), that the experimental presentations were done correctly ("experimenter error") and that subjects were released by their handlers using the correct method and at the correct time ("handler error"). See below for numbers and details regarding reliability coding for attention, experimenter error and handler error.

Second, coders recorded subjects' approach behavior. A subject was considered to have approached when he/she had two front paws in or on the approach box (see Fig 1). Measures of approach behavior were recorded as the absolute number of approaches during the baseline half of the session and during the test half of the session.

Third, coders recorded subjects' reaction time on each trial. Reaction time was coded as the duration of time between the time the handler released the subject and the time the subject first entered the approach box. Additionally, to ensure that trials were roughly the same duration, the entire trial interval was also coded (the time between the subject's release and the time the experimenter turned to return the subject to the handler).

Finally, coders watched for two types of subject referencing behavior: references towards the experimenter ("subject-to-experimenter": did the dogs head orient upward toward the experimenter?) and references towards the subject handler ("subject-to-handler": did the dog orient its head toward the handler?). References were recorded as a binary present/absent variable and were recorded only if they occurred within the trial interval.

A random subset of approximately 10% of sessions was independently re-coded for reliability (dogs: 16 sessions, 11.1% of total, 482 trials; dingoes: 6 sessions, 10.9% of total, 177 trials). Reliability on all categorical variables was high (attention: dogs = 99.6%, dingoes = 98.3%; experimenter error: dogs = 99.8%, dingoes = 100%; handler error: dogs = 98.7%, dingoes = 98.9%; approach: dogs = 100%, dingoes = 98.3%; subject-to-experimenter references: dogs = 89.2%, dingoes = 89.8%, subject-to-handler references: dogs = 97.7%, dingoes = 91.0%). Reaction time was also coded reliably: coders' times were highly correlated (dogs: Pearson's correlation, $r(383) = 0.98$, $P < 0.001$; dingoes: Pearson's correlation, $r(123) = 0.99$, $P < 0.001$).

## Analyses

All statistical analyses were conducted with R statistical software (Version 3.6.3, R Foundation for Statistical Computing, 2020). Approach and referencing behavior were analyzed using Generalized Linear Mixed Models (GLMMs) with binary response terms (1 = approach/reference, 0 = no approach/no reference). Due to design differences between dog and dingo testing, predictor variables differed across analyses. In dog analyses, predictors of interest were Session Half (Baseline vs. Test), Session Type (Inequity vs. Control), Control Group (SNR, NSC1, NSC2, NSNR), Session Order (was Inequity or control conducted first?) and the interaction

between Session Type and Control Group. In dingo analyses, predictors were Session Half (Baseline vs. Test) and Condition (Inequity, SNR, NSC1, NSC2, NSNR). Reaction time data were log transformed and analyzed using Linear Mixed Models (LMMs). Mixed models were run using package lme4 [50].

In all mixed models, subject identity was fit as a random effect (intercepts) to control for repeated measures. All analyses began with a null model, which included only 'subject ID' as an explanatory variable to test how much variation in each dependent variable could be accounted for by individual variation. Following this, a full model was created, which included all predictors of interest. Full models were compared to null models using likelihood ratio tests (LRTs) to determine whether including predictors provided a better fit to the data than simply including ID. A minimal model was then created from the full model by sequentially dropping terms in the model to assess how much their inclusion improved model fit (using LRTs).

### Ethical note

This work was approved by Harvard University IACUC Protocol 28–25. Dog guardians provided informed consent by signing a consent form before their dog participated. Permission to work with dingoes was granted by the Dingo Discovery Sanctuary and Research Centre Director. The identifiable individual depicted in the center of Fig 1 has given written informed consent (as outlined in PLOS consent form) to be in this image.

## Results

### Species differences

Before investigating condition effects on dog and dingo behavior, overall differences between the species on approaches, reaction time and referencing were examined (Fig 2). When collapsing across conditions, dogs were more likely to approach overall compared to dingoes (Fig 2A; GLMM predicting approach as a function of Species: $\chi^2_1 = 9.38$, $P = 0.002$; Table 2). Dogs also tended to approach faster than dingoes (Fig 2B; LMM predicting logged reaction time as a function of species: $\chi^2_1 = 6.81$, $P = 0.009$; Table 2). Finally, compared to dingoes, dogs showed higher levels of both subject-to-experimenter referencing (Fig 2C; GLMM predicting subject-to-experimenter referencing as a function of Species: $\chi^2_1 = 20.93$, $P < 0.001$; Table 2) and subject-to-handler referencing (Fig 2D; GLMM predicting subject-to-handler referencing as a function of Species: $\chi^2_1 = 4.23$, $P = 0.04$; Table 2). Taken together, these results suggest that dogs were more motivated to approach and more attentive to humans than dingoes. Given these differences, all subsequent analyses were performed separately for dogs and dingoes in order to investigate the relative effects of experimental condition within each species.

### Approaches

**Domestic dogs.** Subjects approached on almost all 15 trials during the baseline half of sessions, when they were receiving a reward (14.82 ± 0.52; S1 Fig in S1 File). In contrast, they approached less in the test half of sessions, when they were not receiving a reward (9.31 ± 4.23 trials). Indeed, Session Half was a significant predictor of approach behavior (GLMM: $\chi^2_1 = 1001.1$, $P < 0.001$). Since dogs almost invariably approached in the baseline, their baseline approach behavior was not analyzed further.

Visual inspection of Fig 3A suggests that, contrary to predictions, dogs were no less likely to approach in the Inequity condition compared to the controls in the test half of sessions. In a GLMM predicting approaches as a function of Control Group, Session Type and their interaction, the two-way interaction between Control Group and Session Type was a significant

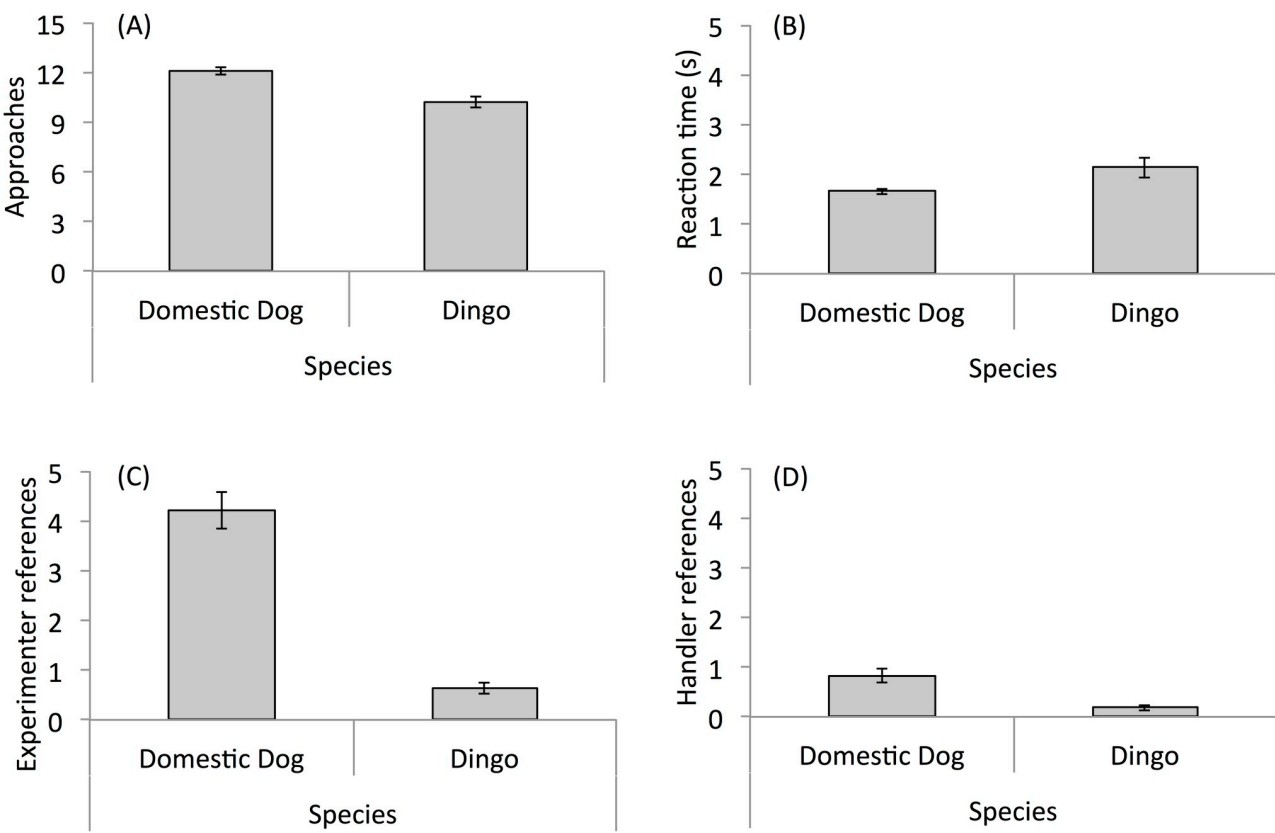

**Fig 2. Bar plots of main dependent measures for dogs and dingoes.** Bar plots show approaches (A), reaction time (B), experimenter references (C) and handler references (D) for domestic dogs and dingoes. Error bars show standard error of the mean.

predictor ($\chi^2_3 = 8.91$, $P = 0.031$). This interaction was due to the fact that dogs in the Non-Social Control 2 group (NSC2; these dogs were tested in both a NSC2 and Inequity session) were less likely to approach during their control session (the completely non-social session; NSC2) compared to their Inequity session, especially when compared to dogs in the SNR control group (SNR; see S1 Fig in S1 File and Table 3 for effects). This finding indicates that social

**Table 2. Estimates and bootstrapped confidence intervals (N = 500 simulations) of fixed effects in mixed models predicting approach behavior, reaction time and referencing behavior in domestic dogs versus dingoes.** Baseline for species was 'dingo'. Table also shows goodness-of-fit statistics.

|  | Approaches | Reaction time | Experimenter references | Handler references |
|---|---|---|---|---|
| Intercept | 1.09 [0.69; 1.51]* | 0.58 [0.42; 0.73]* | -3.22 [-4.01; -2.41]* | -4.79 [-6.04; -3.79]* |
| Species | 0.76 [0.28; 1.23]* | -0.22 [-0.38; -0.05]* | 2.05 [1.17; 2.94]* | 1.17 [0.24; 2.46]* |
| AIC | 5131.71 | 3109.24 | 4847.60 | 1794.74 |
| BIC | 5151.60 | 3134.95 | 4867.49 | 1814.64 |
| Log Likelihood | -2562.85 | -1550.62 | -2420.80 | -894.37 |
| Trials | 5611 | 4575 | 5608 | 5606 |
| Subjects | 83 | 83 | 83 | 83 |
| Variance: Subject | 0.50 | 0.07 | 1.43 | 2.05 |
| Variance: Residual |  | 0.11 |  |  |

* 0 outside the confidence interval

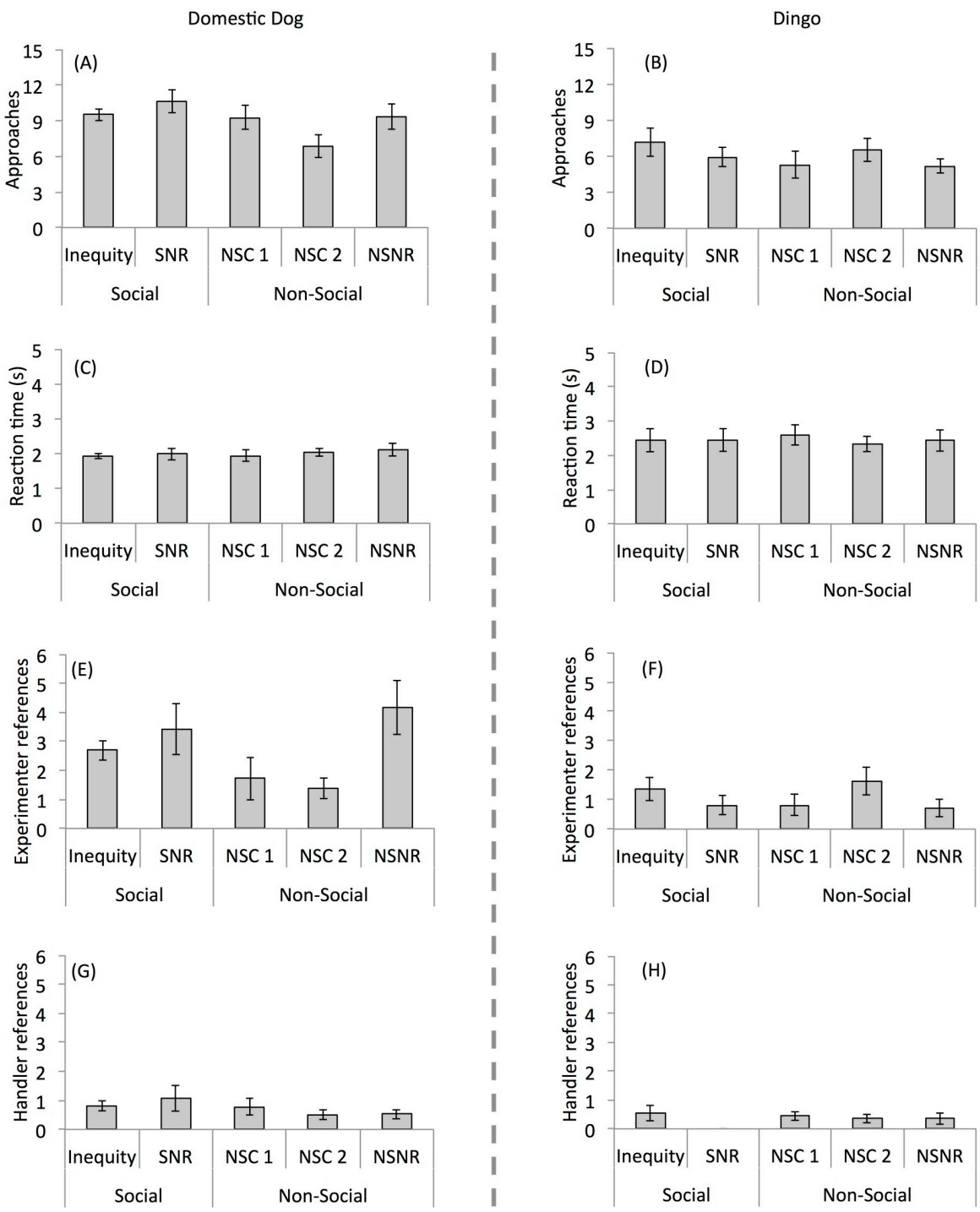

**Fig 3. Bar plots showing test session measures.** Bar plots show approaches (A, B), reaction time (C, D), experimenter references (E, F) and handler references (G, H) for domestic dogs (left column) and dingoes (right column) across conditions. Conditions were as follows: Inequity, subject not rewarded while partner received a reward; Social No Reward (SNR), neither subject nor partner rewarded; Non-Social Control 1 (NSC1), subject not rewarded and partner absent but reward delivered to partner's empty plate; Non-Social Control 2 (NSC2), identical to NSC1 but with a non-social baseline; Non-Social No Reward (NSNR), partner absent and no food distributed on plates. Error bars show standard error of the mean.

**Table 3. Estimates and bootstrapped confidence intervals (N = 500 simulations) of fixed effects in minimal mixed models predicting approach behavior, reaction time and referencing behavior.** Domestic dogs were tested in a between-subject design in which each dog received one session where condition was Inequity and one session where condition was one of four controls. Domestic dog models predicted behavior as a function of Control Group (SNR, NSC1, NSC2, NSNR; baseline = NSC2) and Session Type (Control vs. Inequity; baseline = Control). Dingoes were tested in a within-subject design in which all subjects received all five conditions (Inequity, SNR, NSC1, NSC2, NSNR). Dingo models predicted behavior as a function of Condition. Table also shows goodness-of-fit statistics.

| | Dog approaches | Dingo approaches | Dog reaction time | Dingo reaction time | Dog experimenter refs | Dingo experimenter refs | Dog handler refs | Dingo handler refs |
|---|---|---|---|---|---|---|---|---|
| Intercept | 0.15 | -0.08 | 0.63* | 0.76* | -2.40* | -2.40* | -3.42* | -3.69* |
| | [-0.44; 0.73] | [-0.32; 0.15] | [0.49; 0.77] | [0.57; 0.94] | [-3.32; -1.78] | [-2.74; -2.09] | [-3.97; -3.04] | [-4.43; -3.11] |
| Session Type: Inequity | 0.42 | | -0.04 | | 0.78* | | | |
| | [-0.00; 0.86] | | [-0.13; 0.03] | | [0.19; 1.51] | | | |
| Control Group: NSNR | 0.80 | | -0.02 | | 1.33* | | | |
| | [-0.00; 1.59] | | [-0.23; 0.16] | | [0.41; 2.38] | | | |
| SNR | 1.24* | | -0.06 | | 1.03* | | | |
| | [0.42; 2.09] | | [-0.25; 0.13] | | [0.19; 2.24] | | | |
| NSC1 | 0.86* | | -0.11 | | -0.05 | | | |
| | [0.01; 1.74] | | [-0.30; 0.08] | | [-1.03; 1.17] | | | |
| Session Type x Control Group: | -0.19 | | -0.07 | | -1.30* | | | |
| Inequity x NSNR | [-0.78; 0.43] | | [-0.17; 0.04] | | [-2.07; -0.53] | | | |
| Inequity x SNR | -0.86* | | 0.02 | | -1.20* | | | |
| | [-1.43; -0.23] | | [-0.09; 0.13] | | [-2.01; -0.51] | | | |
| Inequity x NSC1 | -0.47 | | 0.14* | | -0.42 | | | |
| | [-1.07; 0.12] | | [0.03; 0.25] | | [-1.23; 0.37] | | | |
| AIC | 2227.42 | 943.80 | 1034.99 | 312.02 | 1724.21 | 404.69 | 784.15 | 175.95 |
| BIC | 2277.65 | 952.85 | 1086.94 | 323.40 | 1774.42 | 413.73 | 795.31 | 185.00 |
| Log Likelihood | -1104.71 | -469.90 | -507.49 | -153.01 | -853.10 | -200.34 | -390.08 | -85.98 |
| Trials | 1960 | 681 | 1333 | 328 | 1957 | 681 | 1957 | 681 |
| Subjects | 72 | 11 | 72 | 11 | 72 | 11 | 72 | 11 |
| Variance: Subject | 1.25 | 0.08 | 0.06 | 0.10 | 1.29 | 0.08 | 1.54 | 0.38 |
| Variance: Residual | | | 0.11 | 0.13 | | | | |

* 0 outside the confidence interval

factors may have influenced dogs' approach behavior, such that dogs were less inclined to approach when tested in a completely non-social session.

**Dingoes.** Dingoes were more likely to approach in the baseline half of sessions (14.35 ± 0.96 trials; S1 Fig in S1 File) than the test half of sessions (6.02 ± 3.13 trials; GLMM: $\chi^2_1 = 505.53$, $P < 0.001$; Fig 3B). Thus, like dogs, dingoes were sensitive to the absence of food and were more likely to approach when they were being rewarded for doing so. However, there were no differences in approaches across the different conditions during the test sessions (GLMM: $\chi^2_4 = 3.83$, $P = 0.4$), demonstrating that dingoes were no less likely to approach during the Inequity condition than during the controls.

## Reaction time

**Dogs.** Dogs approached more slowly on the test half than the baseline half of sessions (LMM predicting logged reaction time as a function of Session Half: $\chi^2_1 = 1188.2$, $P < 0.001$), indicating that they were less motivated to approach when doing so did not result in a treat. Contrary to predictions, dogs were not slower in the inequity condition compared to the control conditions (Fig 3C; S2 Fig in S1 File). There was, however, a significant interaction

between Control Group and Session Type (LMM: $\chi^2_3 = 16.99$, $P < 0.001$). This was a weak effect (Table 3), which appears to be due to the fact that dogs in the NSC2 group showed more consistency in reaction time across session types (Inequity vs. Control) than dogs in other groups, especially those in the NSC1 group.

**Dingoes.** Dingoes approached more slowly in the test half of sessions than in the baseline (LMM predicting logged reaction time as a function of Session Half: $\chi^2_1 = 116.32$; $P < 0.001$) but there were no condition effects on dingoes' reaction times during test trials (LMM predicting logged reaction time as a function of Condition: $\chi^2_4 = 2.79$, $P = 0.59$; Fig 3D). Thus, like dogs, dingoes approached more quickly when they were being fed than when they were not being fed and did so irrespective of the context in which inequity was presented.

### Human referencing behavior

**Dogs.** Dogs looked more at the experimenter during the baseline half of sessions than during the test half of sessions (GLMM predicting experimenter referencing (binary) as a function of Session Half: $\chi^2_1 = 227.62$, $P < 0.001$). An examination of experimenter referencing in the test half of sessions showed that dogs looked less at the experimenter during two of the test sessions where the partner was absent: there was a significant interaction between Control Group and Session Type (GLMM: $\chi^2_3 = 17.02$, $P < 0.001$; Table 3). This interaction was due to the finding that dogs tested in Non-Social Control 1 and 2 groups (NSC1 and NSC2) tended to look at the experimenter less in their control session than their Inequity session while dogs in the other control groups did not show this difference (S3E Fig in S1 File).

An investigation of dogs' handler referencing behavior showed that handler references did not vary by Session Half (GLMM predicting handler referencing (binary) as a function of Session Half: $\chi^2_1 = 0.13$, $P = 0.718$). Dogs rarely looked back at their handlers during the test half of sessions (Fig 3G) and their tendency to look back was unaffected by Session Type, Control Group, Session Order or the interaction between Session Type and Control Group ($P$s > 0.3).

**Dingoes.** Overall rates of experimenter referencing were low in dingoes (Fig 3F). Dingoes looked at the experimenter more often during the test than the baseline half of sessions (GLMM predicting experimenter referencing (binary) as a function of Session Half: $\chi^2_1 = 47.31$, $P < 0.001$). However, dingoes did not differ in their experimenter referencing behavior across conditions during test trials (GLMM; $\chi^2_4 = 5.39$, $P = 0.249$; Fig 3F). Similarly, dingoes were more likely to look at handlers during the test half of sessions compared to the baseline (GLMM predicting handler referencing (binary) as a function of Session Half: $\chi^2_1 = 24.32$, $P < 0.001$), though this behavior was infrequent (Fig 3H). There were no strong differences in their handler referencing behavior across conditions in the test half of sessions (Fig 3H), though the model showed a marginal effect of condition (GLMM: $\chi^2_4 = 8.74$, $P = 0.068$), most likely due to the fact that handler references were particularly low in the SNR control.

### Discussion

Neither dogs nor dingoes showed behavior consistent with inequity aversion in this study. Contrary to the prediction that subjects would be least motivated to approach in the inequity condition, neither dogs nor dingoes were sensitive to the absence of a reward *specifically* when their partner was being rewarded. Additionally, there were no systematic differences in dogs' and dingoes' approach latencies or referencing behavior in the inequity condition relative to controls.

Comparative psychology, like psychology more generally, suffers from a replication problem [51, 52]. This problem can be attributed to myriad features of comparative work, including difficulties associated with working with different species, small sample sizes and the fact that

comparative work can be expensive. Work with dogs is useful in that it helps address some of these difficulties because large numbers of dogs are present in most cities that house comparative cognition researcher groups and there are fewer expenses associated with research with dogs than with many other species [51]. This makes it somewhat easier to attempt replications —exact and conceptual—of dog findings. Indeed, some groups have even involved 'citizen science' in dog research, which may help replicate different effects in a very large sample of dogs [53]. In this vein, the present study represents a useful contribution to the comparative cognition literature in that it sought to examine inequity aversion in dogs, a species known to show this response in certain contexts (reviewed in [23]) and additionally extended this line of work by testing inequity aversion in a new species, the dingo, as well as in a new experimental context.

In the present study, subjects gradually ceased participation in the task regardless of experimental condition meaning that, although dogs and dingoes did not show a pattern of behavior consistent with inequity aversion, subjects in both species *did* show sensitivity to the presence of a reward. Dog and dingo subjects showed a higher level of motivation to approach when they were being rewarded for doing so: they approached more overall and more quickly in the baseline half of sessions, in which both subject and partner were rewarded for approaching the plate, than in the test half of sessions, in which subjects were never rewarded. Subjects' reluctance to approach when they were not rewarded can of course be explained by the extinction of a response that no longer results in food. However, the present study further indicates that this extinction response was not affected by the presence of a partner receiving a relatively better deal. As discussed in the Introduction, subjects were predicted to stop approaching in the absence of food but they were predicted to stop *earlier* in the Inequity condition than in the controls. These results clearly indicate that subjects' behavior did not conform to this prediction. That being said, the discussion below highlights reasons why the task employed in the current study may not be an ideal paradigm for examining inequity aversion in canids.

Comparisons between dog and dingo subjects revealed a number of behavioral differences between the two species: dogs approached more, approached more quickly and showed more social referencing compared to dingoes. These findings align with past work showing that domestic dogs are more motivated to interact with humans than dingoes ([54, 55] versus [56]; but see [57]). Furthermore, the observed difference in human referencing behavior is consistent with work that has compared referencing between dogs and dingoes [56, 58] and other canids [49]: non-domesticated canids were less likely than dogs to turn to humans when faced with an unsolvable problem. Thus, data from this study add to the comparative canid literature by replicating existing results showing that non-domesticated canid species are less likely to socially reference humans than are domestic dogs.

Despite the finding that neither dogs nor dingoes in the present study showed inequity aversion, there were some hints that at least dogs' behavior was affected by the presence of a partner. For example, dogs were least likely to approach in the Non-Social Control 2 condition, which was an entirely non-social session (i.e., partner was absent for both the baseline and test trials). Additionally, dogs looked at the experimenter less during test trials in NSC1 and NSC2 than in the other controls or Inequity sessions. A possible explanation for this finding is that, in some cases, the presence of a social partner increased dogs' motivation to approach. However, this suggestion is made tentatively because consistent effects across the non-social controls were not observed.

While dogs did look at the experimenter moderately often, there was no pattern in their referencing behavior across experimental conditions. Thus, at present there is no evidence that dogs treat social inequity—inequality presented between the subject and a social partner—differently than inequity presented in non-social contexts—inequality presented when no social

partner is present. However, examination of subject-to-experimenter referencing behavior revealed an unexpected difference between dogs and dingoes: dogs referenced the experimenter relatively more when they were being rewarded than when they were not being rewarded whereas dingoes showed the opposite pattern. This was unexpected because past work found that dogs reference humans when they are unable to solve a problem ([49], but see [59], relevant studies reviewed in [60]). It therefore seems surprising that dogs did not attempt to engage the human experimenter more in the conditions where food was not being put on the plate. It is possible that dogs in this task did not understand the human experimenter's role in the experiment or were not viewing the absence of food as a social problem. Alternatively, because subjects had only a short time to approach, they may have spent most of the test trial intervals searching for absent food, which would have decreased opportunities for human referencing.

The possibility that dogs may not have understood the role of the experimenter and/or viewed the absence of food as a social problem is worth considering beyond the context of subject-to-experimenter referencing. In this vein, one potential limitation of the present design is that I do not have clear evidence that subjects were paying attention to the food that experimenters placed on the partner's side of the barrier. Food was presented *in view* of the subject but this does not equate with attention. The reason for simultaneous food presentation was to bring this task into alignment with other work on inequity aversion, particularly work with humans, in which participants have the option to accept or reject allocations of rewards with knowledge of the full allocation. This is in contrast to much of the work on inequity aversion in animals, in which a partner receives a reward and then the subject decides to accept or reject what they are offered. In the present task, simultaneous presentation may have resulted in subjects attending only to the food presented on their side. Given this, I cannot rule out the possibility that subjects paid attention only to the food on their side of the barrier, thus making the social aspect of the interaction less salient. For similar reasons, I cannot rule out the possibility that subjects largely ignored the role of the experimenter in generating unequal outcomes (see [61] for a study highlighting the important role experimenters play in inequity aversion tasks with nonhuman animals). However, the finding that both dogs and dingoes looked at experimenters across conditions suggests that they were attending to experimenters at least to some extent. Nevertheless, future work using this task could first place food on the partner's side and then on the subject's side. This methodological change may help encourage subjects to attend to the food on the partner's side as well as to the experimenter as the source of inequalities, thereby increasing the social salience of the task.

How do these results fit in with other work on inequity aversion in canids? A growing body of work has demonstrated that dogs show inequity aversion in a 'give paw' task [25, 27, 29] and one study has shown that pack-living dogs and wolves show inequity aversion in a 'buzzer press' task [31]. Why, then, did subjects tested in the current study not show a similar response to inequity? While future work is needed to answer this question properly, there are at least two non-mutually exclusive possible explanations that warrant consideration.

One candidate explanation for the difference between these finding and findings from past work is that previous studies using the 'give paw' task tested dogs with familiar partners whereas the present study only tested unfamiliar dog partners. The decision to pair unfamiliar dogs offered two main benefits. First, research on humans and chimpanzees suggests that individuals in a close relationship are likely to be more tolerant of inequity than unfamiliar individuals [15, 62, 63]. Second, most research with humans focuses on participant pairs who do not know one another and thus this design helps align the canid inequity aversion work with human work, potentially helping to draw conclusions across the two species (see work on so-called 'up-' and 'down-linkage' tasks; [64, 65]). However, the decision to pair subjects with an

unfamiliar partner may have importantly affected the nature of the task. It is reasonable to predict that dogs may be selectively sensitive to how familiar individuals are treated relative to themselves, perhaps because they are particularly used to paying attention to familiar partners. Indeed, recent work suggests that it is important to consider subjects' attention in social situations [66] and that aspects such as familiarity [67] and relationship [68] influence social attention in dogs. Relatedly, pairing unfamiliar subjects may have introduced a level of distraction into the task that is absent from other inequity work with social canids. Finally, one possible reason why inequity aversion in dogs may be restricted to familiar pairs relies on a relatively simple associative account. If two dogs live in a household with a human who routinely feeds them at the same time, it would be relatively easy for an individual to learn a simple rule of thumb: when a partner gets food from a human, expect to get food from a human. By this account, a negative reaction to inequity in familiar pairs is a reaction to a violation of this learned association. Future work could attempt to test this possibility by differentially training the association of "equality" in two groups of dogs and then comparing both groups' performance on an inequity aversion task.

A second possible explanation for the lack of inequity aversion in this present study is that it employed a task that did not require specific training, unlike the 'give paw' and 'buzzer press' tasks. This decision had the benefit of making the task more inclusive for both dogs and dingoes because it relied on a behavior already part of all individuals' behavioral repertoires (i.e., approach). However, this task may not have involved enough 'effort' on the part of participants so as to trigger an inequity aversion response. That is, canids may be specifically sensitive to scenarios in which they perceive differential pay to be allocated for equal effort. Consistent with this hypothesis, results from Horowitz's [30] study, which also included approach as the target response as opposed to a more 'effortful' response, did not find results consistent with inequity aversion. However, work on inequity aversion in rats [*Rattus norvegicus*; 69] used an approach task and did find evidence for inequity aversion. Thus, the importance of effort in inequity aversion tasks in nonhuman animals deserves more attention and its role may well vary across species. Regardless, future work addressing the role of effort in eliciting inequity aversion in dogs and dingoes and testing whether such a response is the result of specific experience will shed important light on the contexts in which inequity aversion is most likely to emerge, helping us better understand its function in canids.

In addition to exploring these two candidate explanations for why inequity aversion was not revealed in this task, it is worth discussing additional features of the study that may have limited its utility in measuring inequity aversion in canids. First, in this task, all subjects were initially run in a baseline half of each session, in which both subject and partner were rewarded (except in the fully non-social condition: NSC2), and then run in an experimental half in which the subject received no reward (and the presence/absence of a partner and reward on the partner's side varied by experimental condition). The logic of this design was that subjects' motivation would decrease regardless of experimental condition but the extent to which it decreased would vary depending on whether the partner was present and being rewarded. However, a possible by-product of this design is that subjects' motivation decreased to the point where they no longer attended to the partner or the partner's reward, thereby diminishing the probability that this task would capture inequity aversion. Second, subjects in this study were paired with only a small number of partners (two confederate partners were involved in the dog sessions and one confederate was involved in the dingo sessions). If familiarity exerts an important influence on inequity aversion, involving unfamiliar partners may have limited the chances that inequity aversion would be revealed in this task. Further, if inequity aversion is particularly likely to emerge with specific *kinds* of partners, the small number of partners involved in this study may have decreased the possibility that inequity aversion

would emerge. And, from an experimental design perspective, involving only a small number of partners represents a problem in terms of pseudo-replication. Future work could consider using a sufficiently large number of partners in the task so as to be able to control for and potentially explore the role of partner identity in analyses. Third and finally, dogs and dingoes in the present task were tested in environments that were at least somewhat familiar to subjects in each group (dogs were tested inside whereas dingoes were tested outside). However, this difference introduces a potential confound that was not accounted for: namely, subjects of each species were not tested in exactly the same environment. This difference may have contributed to some of the species-level differences. For instance, if being tested outside is more distracting, this feature of the testing set-up may have contributed to the finding that social referencing was generally more common in dogs than in dingoes. These are the types of design trade-offs that are often involved in cross-species work where it is important to consider what is necessary to standardize and what is necessary to calibrate to species-specific needs but they nevertheless are important to examine when drawing inferences from cross-species differences.

In sum, the present study contributes data from a new species and a new task to the growing literature on inequity aversion in canids. Results from this study revealed interesting behavioral differences between dogs and dingoes, differences that are broadly consistent with other work comparing these two species [42, 56]. However, findings from this study show no evidence for inequity aversion in either dogs or non-domesticated dingoes. Given existing work that has repeatedly demonstrated inequity aversion in the 'give paw' task, an important avenue for future work is to understand why it emerges in the 'give paw' context and not others [28, 30], including the context examined here. Fortunately, strides are already being made to help explain which experimental features support the expression of inequity aversion in dogs [32], a pursuit that will undoubtedly add value to this area of comparative cognition: a better understanding of the taxonomic distribution of inequity aversion in social canids as well as the experimental contexts in which it emerges can shed important light on the selective forces that shaped inequity aversion in canids as well as in other species, including our own.

## Supporting information

**S1 File.**
(PDF)

**S1 Data.**
(CSV)

**S2 Data.**
(CSV)

**S3 Data.**
(CSV)

**S4 Data.**
(R)

## Acknowledgments

I am grateful to Lynn and Peter Watson for allowing me to work at the Dingo Discovery and Research Centre. I thank Kylie Venardos, Erin Washington, Lyn Whitworth, Winston Chang, Jillian Jordan, George Karas, Gordon Kraft-Todd, Brooke Rice, Erin Washington and Owen Wurzbacher for help with this study. I am especially grateful to Lindsey Powell and Heather Shattuck-Heidorn for allowing their dogs to be tested as confederate partners. Thanks also to

Jordan Comins and Marc Hauser for their help designing this study and to Richard Wrangham, Alex Thornton, Brad Smith, Laurie Santos, Angie Johnston and Mikey Bogese for comments on an earlier version of this manuscript.

## Author Contributions

**Data curation:** Katherine McAuliffe.

**Formal analysis:** Katherine McAuliffe.

**Funding acquisition:** Katherine McAuliffe.

**Investigation:** Katherine McAuliffe.

**Methodology:** Katherine McAuliffe.

**Project administration:** Katherine McAuliffe.

**Resources:** Katherine McAuliffe.

**Writing – original draft:** Katherine McAuliffe.

**Writing – review & editing:** Katherine McAuliffe.

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
