## [Decision Letter · Decision Letter 0]

5 Jan 2021

PONE-D-20-38064

A comparative test of inequity aversion in domestic dogs (Canis familiaris) and dingoes (Canis dingo)

PLOS ONE

Dear Dr. McAuliffe,

Thank you for submitting your manuscript to PLOS ONE. After careful consideration, we feel that it has merit but does not fully meet PLOS ONE’s publication criteria as it currently stands. Therefore, we invite you to submit a revised version of the manuscript that addresses the points raised during the review process.

Many thanks for submitting your manuscript to PLOS One

It was reviewed by two experts in the field, and they have recommended some revisions be made prior to acceptance

I therefore invite you to make the modifications as suggested, and write a response to reviewers which will expedite revision upon resubmission

I wish you the best of luck with your revisions

Hope you are keeping safe and well in these difficult times

We look forward to receiving your revised manuscript.

Kind regards,

Simon Clegg, PhD

Academic Editor

PLOS ONE

2. In your Methods section, please provide additional details regarding participant consent from the owners of the animals.

In the ethics statement in the Methods and online submission information, please ensure that you have specified (i) whether consent was informed and (ii) what type you obtained (for instance, written or verbal).

If the need for consent was waived by the ethics committee, please include this information.

5. We note that Figure 1 includes images of participants in the study. 

As per the PLOS ONE policy (http://journals.plos.org/plosone/s/submission-guidelines#loc-human-subjects-research) on papers that include identifying, or potentially identifying, information, the individual(s) or parent(s)/guardian(s) must be informed of the terms of the PLOS open-access (CC-BY) license and provide specific permission for publication of these details under the terms of this license.

a. Please download the Consent Form for Publication in a PLOS Journal (http://journals.plos.org/plosone/s/file?id=8ce6/plos-consent-form-english.pdf). The signed consent form should not be submitted with the manuscript, but should be securely filed in the individual's case notes.

b. Please amend the methods section and ethics statement of the manuscript to explicitly state that the patient/participant has provided consent for publication: “The individual in this manuscript has given written informed consent (as outlined in PLOS consent form) to publish these case details”.

If you are unable to obtain consent from the subject of the photograph, you will need to remove the figure and any other textual identifying information or case descriptions for these individuals.

Reviewers' comments:

Reviewer's Responses to Questions

**Comments to the Author**

1. Is the manuscript technically sound, and do the data support the conclusions?

Reviewer #1: Yes

Reviewer #2: Yes

2. Has the statistical analysis been performed appropriately and rigorously? 

Reviewer #1: Yes

Reviewer #2: Yes

3. Have the authors made all data underlying the findings in their manuscript fully available?

Reviewer #1: Yes

Reviewer #2: Yes

4. Is the manuscript presented in an intelligible fashion and written in standard English?

Reviewer #1: Yes

Reviewer #2: Yes

5. Review Comments to the Author

Reviewer #1: This was a very clearly written and well balanced article. I found it very easy to read and follow and I especially enjoyed the inclusion of dingoes and the comparisons drawn with domestic dogs. I think this article will be well received and I have only a few minor questions/comments for clarity.

I believe this is a single author article and yet sometimes the third person voice is used

(e.g., line 179 “Our third measure…”) – I suggest just using first person singular

Line 267 “ Regardless of condition, subjects never received a treat for approaching during test trials.” Approaching what?

Line 268 “Test trials were always preceded by baseline trials in order to control for the possibility that subjects would lose motivation over the course of a session: because the non-rewarded part of the session always occurred after the rewarded part of the session, the potential effects of loss of motivation were consistent across all conditions.” But do you think this may have also buffered any effects of inequity?

Table 1 please state explicitly in each of the descriptions what and how many rewards were put on each plate in each condition. If the reward count was consistent across conditions, perhaps this can go in the caption.

Table 1 in the non-social baseline, was food also put on the second plate?

Line 328 Sorry if I missed it, but how far were the dogs and dingoes (in distance) from the plates at the start of each trial?

Line 348 “In cases where a trial was deemed invalid during testing (e.g., due to a mistimed dog release by the handler) the trial was redone.” How often did this happen?

Line 344 what was the inter-trial interval? i.e how long did the animals have to eat the food (if available) before being recalled to their handler?

Line 413 this ethical approval reference is a little unclear – what institution is this associated with? And does this apply to both the USA and Australia? Furthermore, please ensure and confirm that your reporting meets the ARRIVE 2.0 guidelines (Percie du Sert et al. 2020 The ARRIVE guidelines 2.0: Updated guidelines for reporting animal research. PLoS Biol 18(7): e3000410).

You might consider including tables S3 and S4 in the main body of your article.

I think you should also consider providing your raw R script/code as supplementary materials to accompany your data sets.

Reviewer #2: Summary:

The current study compares dogs’ and dingoes’ response to inequity in a novel inequity task. Subjects’ approach to a plate was assessed in an inequity condition (i.e. when no food was placed on their plate but food was placed on the plate of a conspecific partner behind a transparent barrier) and compared with responses in control conditions that varied with regards to the presence of a partner and the presence of food on the partner’s plate. If subjects were inequity averse, they should have approached fewer times and taken longer to approach in the inequity than in the control conditions. Overall, neither dogs nor dingoes approached fewer times or took longer to approach in the inequity condition compared with the control conditions. Although the findings in this study contradict those of other studies that demonstrate inequity aversion in dogs, the author points out some potential methodological reasons for the absence of inequity aversion.

Overall, the manuscript is well written and provides a good background to the field. The author also does a good and fair job of interpreting the results in the discussion, particularly in the light of the sizeable literature on inequity aversion in dogs that already exists and potential methodological explanations for the lack of inequity aversion. I think it is also commendable that the author tries to develop a new inequity task for use with non-human animals/dogs that is more similar to tasks used with humans.

I have some comments which I believe will help improve the manuscript and these largely relate to additional points I feel should be raised in the discussion, some of these highlighting methodological issues.

Major comments:

Lines 179 – 183: Although it might make sense that dogs would have a greater tendency to look at the humans, it is not clear enough to me from the introduction so far why this measure is something worthwhile to look at with regards to inequity aversion…e.g. is it a measure of inequity aversion? If there is no difference in social referencing across conditions but there is a difference in approach would this change anything about the conclusion? If there is a difference in social referencing across conditions, it is also not clear what this would mean. It seems to me that the social reference measure would not tell us much. There could be a good reason for including it and there may be a good argument for it but then I would suggest giving a stronger reason here in the introduction. Otherwise, I think the other two measures are sufficient as they give clearer answers to the questions addressed. Whether the social referencing is omitted or explained better or not changed at all is up to the author and should not change the decision regarding acceptance in my view, though excluding the referencing would lead to quite a few changes to the discussion.

Line 207: The fact that there were only 3 partners in the whole study, as far as I understood, is a problem in terms of pseudoreplication. The study potentially asks whether the individuals are averse to inequity with that specific partner. Of course, we do not know whether the unfamiliarity of the partner, or the partner-specific traits, account for much of the response; therefore, it is difficult to assess the extent to which this issue explains the results. There is not much that can be done about this now but I think this point should be raised in the discussion.

Lines 236 – 242: It is good that both species were tested in the type of environment that they were familiar with but the fact that these were different environments could also be a problem. I think this should also be addressed at least briefly in the discussion.

Discussion: In general, the discussion addresses a lot of very good points. One major point which is not really touched on which I would like to see discussed is whether the subjects actually paid attention to the partner and whatever else was happening on the other side of the barrier, including the differences in rewards. The results seem consistent with the idea that the dogs did not pay any attention to what was going on (apart from the indication that the presence of a partner affected some aspects of their behaviour, but one could respond naturally to the presence of a conspecific in the vicinity without paying attention specifically to the events on the other side of the transparent barrier). It would be useful to have an extra variable such as gazing at the partner or gazing at the partner’s food, though I recognise this could be quite difficult to code and might not give a very clear answer. Either way, I think it would be good to include more discussion around this point about attention to the partner and the partner’s rewards. See also related comments about specific points in the discussion.

Lines 586 – 588: I would like to see an elaboration of these points: Why would the subjects not have understood the experimenter's role and why would they not have viewed the absence of food as a social problem? I understand these points are being raised in the context of dogs referencing the human but the answers to these questions might also provide an explanation for why no inequity aversion was observed (if dogs and dingoes are actually inequity averse).

Lines 589 – 591: It seems this could also be an elaboration of or explanation for the points in the previous sentence, even though it is not presented as such i.e. the subjects might not have understood or paid attention to the experimenter as they were focused on searching for food. Also, they might not have viewed it as a social issue as they may have been too focused on whether there was food on their own side. Consider including this as a possible explanation.

Minor comments:

Line 85-86: This should read the “former of which” should it not, as cooperative breeds would have been hypothesized to have a stronger response to inequity?

Line 106: Should this read “…a relatively ¬lower payoff…” or something along these lines.

Lines 129 – 131: It seems as though the opening line of this paragraph is rehashing a point that has already been made. It is confusing on first reading as to why this point is brought up again as it has just been made already. It takes a while before it is clear what the author is getting at in the paragraph. Perhaps beginning this sentence with “although” or something similar, and restructuring the sentence accordingly would be helpful here.

Lines 131 – 132: I find this sentence confusing: “However, past work also points to questions whose answers could help further bolster this claim.” I would consider revising and simplify it.

Line 137 and line 158: If the author wishes, the following reference could also be considered here: Oberliessen et al. 2016 Animal Behaviour ("Inequity aversion in rats"); it is not the same method as this study, but there are similarities particularly with the Horowitz 2012 dog study, as the subject had to make a choice related to the distribution of rewards between itself and a partner.

Lines 139 – 140: This line seems to be repetition of some of the information in lines 135 – 136. I understand from reading on that the author is setting up (in lines 135 onward) what will then be covered in the next 2 paragraphs but it does still come across as repetition. I would consider revising this slightly. Perhaps for example, lines 135-136 could state that the study compares dogs with a non-domesticated social canid….or in lines 139 onward beginning the paragraph with the assumption that the reader knows that Dingoes were tested in an inequity task?

Line 152: Similar to the issue of repetition raised above, the point that “a novel task that requires no specific training” was already made, and was clear.

Line 206: The use of the word confederate here and throughout the manuscript is unnecessary is it not? Wouldn't the word partner be sufficient?

Line 332: What happened with the food if it was not placed on the plate? Did the experimenter hold it in their hand or place it in a pouch or something and if so, could the subject see this? Please clarify in the manucript

Lines 374 – 375: I do not understand the purpose of this sentence: “A video recording of each session was watched and analysed by a research assistant.” The previous 4 paragraphs make it clear that the videos were analysed by research assistants. Perhaps this sentence refers to the reliability coding in which case I recommend combining it with the next sentence or rewording to make it clear that the reliability coding is being referred to here.

Line 377: To what does attention refer in this case?

Line 413: What is Protocol 28-25? Please provide a brief explanation e.g. is this from a university or government.

Line 420 – 424: There is no information given in the methods section on the structure of the models used for this analysis. Please include this.

Lines 442 – 443: What analysis is this conclusion based on specifically? Is this based on a non-significant main effect of the predictor session type or based on visual inspection of plots?

Lines 579 – 580: What is meant here with inequity in non-social contexts? Does it refer to simply having two batches of food (or some other resource) that differ in quality or quantity? Please clarify this in the text.

Lines 583 – 585: Please include a reference for dogs’ referencing of humans when trying to solve a problem. See also Lazzaroni et al. 2020 Animal Cognition (why do dogs look back at a human in an impossible task); in sum, the results of this study suggest that much of dogs’ looking to humans when attempting to solve difficult/impossible problems is simply a consequence of giving up with the task and looking at a salient stumulus.

Line 598 – 599: The wording of this sentence seems to suggest that whether the partners in the previous studies were familiar or unfamiliar is an unknown. As this is not the intended meaning of the sentence, as far as I understand, consider rewording slightly to make it is clear that it is a possibility that the difference in use of the familiar vs unfamiliar partners between the studies is the reason for a difference in results.

Line 609: Is there any research showing that dogs’ attention differs based on familiarity? A reference would be particularly helpful here to support the point. See Horn et al. 2013 Animal Cognition ("Dogs’ attention towards humans depends on their relationship, not only on social familiarity"); see also Range et al. 2009 Animal Cognition ("Social attention in keas, dogs, and human children"). But any insight from any species on the relationship between attention and familiarity would also be helpful here as a reference.

CSV files: It seems that there were different experimenters. It would be good to state this in the manuscript, including how many experimenters and whether the experimenter remained the same for a subject across conditions.

Miscellaneous:

Line 39: The use of square vs. round brackets differs from line 38.

Line 66: Brosnan and de Waal should be the possessive i.e. Brosnan and de Waal’s

Line 556: This should read “current study” rather than “currents study”.

Table S4: Check use of big “X” for one of the interactions but small “x” for others.

Fig. S1, Fig. S2, and Fig. S3 legends: Here it states “…lines extending from boxes indicate minimum and maximum values and black circles indicate outliers”. The lines extending from the boxes do not seem to indicate the min. and max. values as there are additional values (the outliers) beyond their range.

6. PLOS authors have the option to publish the peer review history of their article (what does this mean?). If published, this will include your full peer review and any attached files.

Reviewer #1: No

---

## [Author Response · Author response to Decision Letter 0]

21 Jul 2021

Please see attached letter to reviewers.

---

## [Editor Report · Decision Letter 1]

27 Jul 2021

A comparative test of inequity aversion in domestic dogs (Canis familiaris) and dingoes (Canis dingo)

PONE-D-20-38064R1

Dear Dr. McAuliffe,

We’re pleased to inform you that your manuscript has been judged scientifically suitable for publication and will be formally accepted for publication once it meets all outstanding technical requirements.

Kind regards,

Simon Clegg, PhD

Academic Editor

PLOS ONE

Additional Editor Comments:

Many thanks for resubmitting your manuscript to PLOS One

As you have addressed all the comments and the manuscript reads well, I have recommended it for publication

You should hear from the Editorial Office shortly.

It was a pleasure working with you and I wish you the best of luck for your future research

Hope you are keeping safe and well in these difficult times

Thanks

Simon

---

## [Editor Report · Acceptance letter]

13 Sep 2021

PONE-D-20-38064R1 

A comparative test of inequity aversion in domestic dogs (*Canis familiaris*) and dingoes (*Canis dingo*) 

Dear Dr. McAuliffe:

I'm pleased to inform you that your manuscript has been deemed suitable for publication in PLOS ONE. Congratulations! Your manuscript is now with our production department. 

Kind regards, 

on behalf of

Dr. Simon Clegg 

Academic Editor

PLOS ONE